# Relationship between Tumor Mutational Burden, PD-L1, Patient Characteristics, and Response to Immune Checkpoint Inhibitors in Head and Neck Squamous Cell Carcinoma

**DOI:** 10.3390/cancers13225733

**Published:** 2021-11-16

**Authors:** Kimberly M. Burcher, Jeffrey W. Lantz, Elena Gavrila, Arianne Abreu, Jack T. Burcher, Andrew T. Faucheux, Amy Xie, Clayton Jackson, Alexander H. Song, Ryan T. Hughes, Thomas Lycan, Paul M. Bunch, Cristina M. Furdui, Umit Topaloglu, Ralph B. D’Agostino, Wei Zhang, Mercedes Porosnicu

**Affiliations:** 1Wake Forest Baptist Medical Center, Winston-Salem, NC 27157, USA; kburcher@wakehealth.edu (K.M.B.); jwlantz@wakehealth.edu (J.W.L.); egavrila@wakehealth.edu (E.G.); afaucheu@wakehealth.edu (A.T.F.); axie@wakehealth.edu (A.X.); cwjackso@wakehealth.edu (C.J.); asong@wakehealth.edu (A.H.S.); ryhughes@wakehealth.edu (R.T.H.); tlycan@wakehealth.edu (T.L.J.); pbunch@wakehealth.edu (P.M.B.); cfurdui@wakehealth.edu (C.M.F.); Umit.Topaloglu@wakehealth.edu (U.T.); rdagosti@wakehealth.edu (R.B.D.J.); wezhang@wakehealth.edu (W.Z.); 2Lewisgale Medical Center, Salem, VA 24153, USA; a_abreu0419@email.campbell.edu; 3Lake Erie College of Medicine, Bradenton, FL 34211, USA; jburcher83447@med.lecom.edu

**Keywords:** HNSCC, TMB, immunotherapy, immune checkpoint inhibitors, PD-L1

## Abstract

**Simple Summary:**

Immunotherapy has prompted a dramatic change in the management of head and neck squamous cell carcinoma (HNSCC), but the percentage of patients benefiting from treatment is limited to 20% or less. The application of precision oncology to HNSCC introduces the potential for the emergence of biomarkers that may predict a response to immunotherapy and assist with the selection of patients that may benefit from treatment with an immune checkpoint inhibitors. In this retrospective study, the results of tumor mutational burden and programmed death ligand-1 measurements from HNSCC tumors were evaluated independently for their associations with demographics, risk factors, disease characteristics, survival, and response to ICI. Results of this study are expected to assist in laying the groundwork for creating a framework in which PD-L1 and TMB coexist with other variables to predict response to ICI on an individual level.

**Abstract:**

Failure to predict response to immunotherapy (IO) limited its benefit in the treatment of head and neck squamous cell cancer (HNSCC) to 20% of patients or less. Biomarkers including tumor mutational burden (TMB) and programmed death ligand-1 (PD-L1) were evaluated as predictors of response to IO, but the results are inconsistent and with a lack of standardization of their methods. In this retrospective study, TMB and PD-L1 were measured by commercially available methodologies and were correlated to demographics, outcome, and response to PD-1 inhibitors. No correlation was found between TMB and PD-L1 levels. High TMB was associated with smoking and laryngeal primaries. PD-L1 was significantly higher in African Americans, patients with earlier stage tumors, nonsmokers, and nonethanol drinkers. Patients with high TMB fared better in univariate and multivariate survival analysis. No correlation was found between PD-L1 expression and prognosis. There was a statistically significant association between PFS and response to IO and TMB. There was no association between response to ICI and PD-L1 in this study, possibly affected by variations in the reporting method. Further studies are needed to characterize the biomarkers for IO in HNSCC, and this study supports further research into the advancement of TMB in prospective studies.

## 1. Introduction

Prognoses in oncology have dramatically improved with the development of immunotherapy (IO), particularly with the advent of immune checkpoint inhibitors (ICI). Antibodies to programmed death-1 (PD-1) receptor and the programmed death ligand-1 (PD-L1) are specific kinds of ICI which function by inhibiting the binding of the programmed death-1 (PD-1) receptor to PD-L1, thus allowing tumor cells to be recognized as “other” and eliminated by a patient’s immune system (Figure 1) [1]. Emerging data indicate patients with HNSCC are likely to benefit from breakthroughs in ICI, presumably due to the high levels of circulating immune cells and high levels of neoantigens within these tumors (Figure 1) [2]. Clinical trials evaluating the response to ICIs in recurrent unresectable and metastatic HNSCC have shown significant improvement in overall survival (OS) when PD-1 inhibitors are utilized alone or in combination with chemotherapy [3,4,5,6,7]. 

Despite these promising developments, fewer than 20% of HNSCC patients respond to treatment with ICI within the FDA approved setting, with the majority of patients displaying primary resistance [8]. With current clinical trials seeking to advance PD-1 ICI to curative settings, improved patient selection as a means of increasing the percentage of responders is critical, urging the development of reliable biomarkers predictive of the response to ICI. Many immune biomarkers including PD-L1 expression, tumor mutational burden (TMB), tumor immune cell infiltration, circulating immune cells, HPV, changes within the microbiome, and certain risk factors such as smoking have been suggested as predictors of HNSCC response to ICI, but the data remain in its infancy, without causative association and often with conflicting results within the literature [9]. Furthermore, there remains a limited understanding of how responses to ICI, prognosis, and each of the proposed biomarkers may be impacted by environmental factors or individual patient characteristics.

As a logical biomarker for the prediction of response to anti-PD1/PD-L1 agents, PD-L1 expression remains the only biomarker studied in prospective clinical trials in HNSCC. Studies place HNSCC amongst the malignancies with the highest frequency of PD-L1 positivity, defined as PD-L1 expression ≥ 1% when measured by either tumor proportion score (TPS) or combined positive score (CPS). It has been estimated that between 57% and 82% of HNSCC patients are PD-L1 positive [5,10,11]. PD-L1 expression is often described as an inexact measurement of response to ICI with levels broadly correlating to response rates; however, it is well known that some PD-L1 negative patients do respond to ICI, and some patients with high PD-L1 levels do not [12].

Given the inconsistencies in results produced by the utilization of PD-L1 to predict response to ICI, many alternative biomarkers have been suggested. The chief prospect amongst these alternatives is TMB, a measure of the total number of coding mutations in a tumor’s genome and reported in number of mutations per mega base (mut/Mb) of DNA sequenced [13]. In theory, higher TMB conveys a higher expression of tumor neoantigens, which elicit an increased antitumor immune response, conferring greater sensitivity to ICI (Figure 1) [9]. This has been proven to be the case in many malignancies, and at times, TMB has been reported to outperform PD-L1 in prediction of response; however, the data are incomplete and occasionally contradictory, and thus no consensus has been reached regarding widespread clinical use [13,14,15,16,17,18,19,20,21,22,23,24,25]. Although TMB has been approved for the selection of patients with cancer for IO treatment independent of tumor type [26], few data are available in HNSCC. Additionally, few studies correlate TMB to demographics or survival of HNSCC patients, and the results of studies that do attempt these correlations remain inconsistent [26,27]. The predictive power of TMB in regard to response to ICI in HNSCC also has yet to be defined. To date, there are no prospective studies regarding the use of tissue TMB as a biomarker to predict response to ICI, but retrospective analyses have successfully correlated high TMB with response to ICIs in HNSCC [9,24,27,28,29]. Similarly, the emerging role of circulating/blood TMB remains undefined, but preliminary results are promising [30,31].

Inconsistency in results is common in the literature regarding biomarkers such as PD-L1 and TMB. This is not exclusive to HNSCC. The variation in the assays used to measure these variables and reporting appear to be important contributors. Standardization efforts are ahead for PD-L1, with FDA approval of PD-L1 IHC 22C3 pharmDx reported as CPS, as a companion diagnostic for pembrolizumab treatment in HNSCC. Validation in clinical research and practice is ongoing. No such efforts have been undertaken for TMB, which has not yet been evaluated in prospective setting in HNSCC. 

Joining the effort to lay a groundwork for the use of PD-L1 and TMB biomarkers to guide protocols regarding the use of PD-L1 ICI in HNSCC, this retrospective study correlates the level of expression of these biomarkers with demographic and outcome data in a dedicated HNSCC population. Additionally, the investigators correlate PD-L1 and TMB expression with response to PD-L1 ICI in the cohort of treated HNSCC patients. The primary objective of this study is to investigate the feasibility of continued pursuits of PD-L1 and TMB in prospective clinical trials in which ICI would be used to treat HNSCC. Importantly, the commercially available methodologies for measurement of the biomarkers were utilized.

## 2. Materials and Methods

This is a single-institution retrospective review of adult patients with HNSCC treated at the Wake Forest Baptist Comprehensive Cancer Center between August 2014 and October 2020 who had tumor tissue submitted for next-generation sequencing (NGS) and/or PD-L1 testing. The Wake Forest School of Medicine Institutional Review Board (IRB00057787) reviewed this study and granted approval. HNSCC patients were required to have had a valid TMB or PD-L1 test to be included in this study. Patients with cutaneous SCC or salivary gland cancers were excluded.

TMB was measured via FoundationOne (F1) testing (Foundation Medicine, Cambridge MA, USA) (F1). PD-L1 was analyzed by the standard, FDA-approved, immunohistochemistry 22C3 pharmDx kit, performed commercially by F1 and, in a small number of patients, by Mayo Clinic laboratories. PD-L1 expression was reported as a tumor proportion score (TPS) until 2019 and by the combined positive score (CPS) thereafter. PD-L1 was analyzed both as a 3-tiered and 2-tiered variable. The 3-tiered PD-L1 variable (3tPD-L1) was divided into categories similar to those in Keynote-048 and consisted of three groups: those with a PD-L1 of 0 (3tPD-L1-(0)), those with a PD-L1 between 1 and 19 (3tPD-L1-(1–19)) and those with PD-L1 greater than or equal to 20 (3tPD-L1-(20+)). Due to the low number of patients in the 3tPD-L1-(0) group, PD-L1 was also analyzed as a 2-tiered variable (2tPD-L1). In the 2tPD-L1, the 3tPD-L1-(0) and 3tPD-L1-(1–19) were grouped together and referred to as 2tPD-L1-(<20) and was compared to a group of patients with PD-L1 values greater than or equal to 20 referred to as 2tPD-L1-(20+). TMB was likewise initially divided into three categories (3-tiered-TMB) with low scores (TMB less than 6 mut/Mb), intermediate scores (TMB greater than or equal to 6 but less than 20 mut/Mb), and high scores group (TMB greater or equal to 20 mut/Mb) as recommended by F1, but, due to poor distribution across the sample population, no analysis was performed for the 3-tiered variable. Similar to PD-L1, TMB was recategorized into a 2-tiered variable, with those with TMB scored less than 6 mut/mb in the TMB-(<6) category and those with TMB of 6 or greater in the TMB-(6+) category. In instances in which PD-L1 or TMB tests were repeated, the highest resulted number was reported.

Demographic data and patient characteristics were obtained from the electronic medical record and included age (greater or less than 60 years old), sex, disease stage at diagnosis per AJCC 8th edition, HPV by PCR or p16 status, smoking status (grouped as never-smokers vs. ever-smokers, where ever-smokers were defined as former or current smokers), alcohol use, tumor subsite (oral cavity, oropharynx, larynx, hypopharynx, nasopharynx, paranasal sinuses, or unknown primary), and treatment received before tumor tissue collection (chemotherapy, radiotherapy, or both).

Outcome measures included OS measured from the time of diagnosis and from the time of tumor tissue collection. Survival at 1 and 2 years measured from the date of tumor tissue collection, survival at the end of the study, and extent/burden of disease at last visit were also included in outcome data. It should be noted that for all calculations in which the extent of disease was measured, three categories were considered. These were defined as “no evidence of disease”, “localized disease”, and “metastatic disease”. Multivariate analysis was performed for both TMB and PD-L1 groups separately.

Treatment response was measured by CT or MRI and categorized according to RECIST v1.1. Patients with complete response (CR), partial response (PR), or stable disease (SD) for at least 6 months as best overall response (BOR) were grouped in a category called “responders”. Patients who progressed through the treatment (PD) without achieving a response as above were called “nonresponders”. Progression-free survival (PFS) was measured from the first day of treatment with ICI to the day of confirmed tumor progression, or to the day of death if the patient died before tumor progression was documented, or until the last visit if there was no tumor progression.

### Statistical Analysis

Descriptive statistics were calculated for all variables. These included means and standard deviations for continuous measures and counts and percentages for categorical measures. TMB and PD-L1 levels were portioned into tiers as described above (TMB into 2 tiers and PD-L1 into 2 or 3 tiers). We then examined the association between categorical variables and the TMB/PD-L1 groupings using Fisher’s exact test (for binary variables) and chi-square test for categorical variables with 3 or more levels. Continuous variables were compared across TMB/PD-L1 groups using t-tests (for 2 tier groups) and one-way analysis of variance models for 3tPD-L1. Time-to-event data were examined in two ways: one examining time from diagnosis until event (i.e., death) and the second examining time from testing until event. Kaplan–Meier curves were generated for examining survival distributions both overall and by TMB or PD-L1 groups. Log-rank tests were used to compare groups. Next, Cox proportional hazards regression models were fit to examine the relationship of TMB or PD-L1 groups with survival after adjusting for patient level characteristics including age, tobacco use, tumor site, stage at diagnosis, and prior treatment with combined chemoradiation therapy. Next, we evaluated treatment response to immunotherapy as BOR and PFS. Fisher’s exact tests and chi-square tests (as described above) were used to determine whether PD-L1 and TMB categories were associated with the BOR treatment categories. We compared average PFS days between responders and nonresponders using a 2-sample t-test. Next, we compared PFS days by PD-L1 and TMB categories as described above using 1-way ANOVA models and PD-L1 and TMB levels (as continuous values) using Pearson correlations. Hazard ratios and corresponding 95% confidence intervals were estimated from these proportional hazard regression models. In all analyses, an alpha level of 0.05 was used to determine the significance of data. Statistical analysis system (SAS) 9.4 was used to perform all analyses in this study.

## 3. Results

### 3.1. Patient Characteristics

In total, 139 patients met inclusion criteria for this study. Of these, 128 patients had TMB results, 95 patients had PD-L1, and 92 patients had results for both metrics. The demographic and disease characteristics of the patients included in this analysis are available for review in Table 1. Age, race, and gender in this study are congruent with a standard population of patients with HNSCC.

### 3.2. Prevalence of PD-L1 and TMB within the Study Population and Correlation between the Two Variables

Of the 95 patients with recorded measurements of PD-L1 expression, 80 patients (84%) had results from testing performed by F1, and 15 patients (16%) had results from testing performed by the Mayo Clinic Laboratory. PD-L1 was measured by TPS in 52 patients (55%) and by CPS in the remaining 43 patients (45%). The mean PD-L1 score was 26.41% (standard deviation 33.78). The median score was 10%. Twelve patients (13%) had 3tPD-L1-(0). In addition, 46 patients (48%) had 3tPD-L1-(1–19), and 37 patients (39%) had 3tPD-L1-(20+) (Figure 2A). The 2tPD-L1 re-distribution resulted in 58 patients (61%) in the 2tPD-L1-(<20) group and 37 patients (39%) in the 2tPD-L1-(20+) group.

The mean TMB of the 128 patients included in TMB analysis was 6.98 mut/Mb (standard deviation 6.85), and the median was 5.0 mut/Mb. The TMB-(<6) category consisted of 70 patients (55%), and 58 patients (45%) were in the TMB-(6+) category (Figure 2B).

A total of 92 patients had both PD-L1 and TMB testing results available. Of these, 12 patients (13%) were in the 3tPD-L1-(0) category, 44 patients (48%) were in the 3tPD-L1-(1–19) category, and 35 patients (39%) were in the 3tPD-L1-(20+) category. In addition, 38 patients (42%) were in the TMB-(6+) category, and 53 patients (58%) were in the TMB-(<6) category. There were no statistically significant correlations identified between PD-L1 expression and TMB as categorical variables (*p* = 0.84) (Figure 2C).

### 3.3. Correlation between PD-L1 Expression and Patient Characteristics

African Americans had statistically significant higher PD-L1 expression than Caucasians, with 69.2% vs. 35.4% classified in the 2tPD-L1 ≥ 20 (*p* = 0.04). The analysis lost statistical significance in the three-tiered variant. There were no significant correlations identified between age or gender and 3tPD-L1 or 2tPD-L1, although it was noted that patients older than 60 and women regardless of age had a tendency toward higher PD-L1 expression. There was no significant correlation between HPV status and PD-L1, both in 2tPD-L1 *(p* = 0.74) and 3tPD-L1 (*p* = 0.35) analysis; however, in the 3tPD-L1 analysis it was noted that there were no HPV-positive patients in the 3tPD-L1-(0) category, while 18.6% of HPV-negative patients were 3tPD-L1-(0) (Table 2).

In the evaluation of modifiable traits, current nonsmokers had a significantly higher percentage of patients in the 3tPD-L1-(20+) category when compared with current smokers (51.7% vs. 33.3%; *p* = 0.04). The same significance held true when never-smokers were compared to ever-smokers (current and former smokers) (*p* = 0.04). Similarly, PD-L1 and alcohol use were associated in a statistically significant manner. More than half of never-drinkers (52.2%) vs. only 25.9% of drinkers were within the 3tPD-L1-(20+) category (*p* = 0.01). The statistical advantage was maintained in the 2tPD-L1 analysis (*p* = 0.04). Low PD-L1 was also noted to be associated with low BMI, but this trend did not reach significance (2tPD-L1 *p* = 0.088 and 3tPD-L1 *p* = 0.07). Exposure to radiotherapy, chemotherapy, or to chemoradiotherapy before tumor tissue collection for NGS or PD-L1 testing did not correlate with either PD-L1 analysis.

There was no correlation found between 2tPD-L1 or 3tPD-L1 expression and the primary tumor site (*p* = 0.36 and *p* = 0.61, respectively), and no trends were identified to influence further analysis.

A statistically significant correlation was found in the 3tPD-L1 analysis of the disease stage (I–IV) at diagnosis, with a higher percentage of patients with early-stage disease having 3tPD-L1-(20+) (58.8% patients with stage I disease vs. 35.6% with stage IV disease; *p* = 0.04). In particular, PD-L1 associated with tumor stage but not with nodal stage. In early tumor stages (T0–T2), 46.5% of patients were identified with 3tPD-L1-(20+) vs. only 32.7% of patients with tumor stages T3–T4, and 0% patients with early T stage vs. 23% of patients with advanced T stages were in the 3tPD-L1-(0) group (*p* < 0.01).

### 3.4. Correlation between TMB and Patient Characteristics

Due to the extremely low proportion of patients within TMB 20+ category (just five patients) in the three-tiered TMB variable (Figure 2B), no attempts were made to a three-tiered TMB correlative analysis. Going forward, all references are limited to the two-tiered TMB analysis. There were no significant correlations found between the TMB and age (*p* = 0.48) or gender. There was a notably higher proportion of women in the TMB-(6+) group (53.8% vs. 41.6%), but the association was not statistically significant (*p* = 0.20). Correlation of TMB with race showed that 48.1% of Caucasians vs. only 23 % of African Americans had a TMB-(6+) (*p* = 0.09). Of note, there was no correlation between TMB and HPV status (*p* = 0.80).

Active tobacco users were significantly more likely to be in the TMB-(6+) category when compared to former and never-smokers (60% vs. 33.3% and vs. 37.8%; *p* = 0.03). There was no significant association between TMB and alcohol use. Patients with a body mass index (BMI) great than 30 were more likely to have low TMB-(<6) (69% vs. 50%), but this trend failed to meet statistical significance (*p* = 0.07). Patients with previous exposure to chemoradiotherapy (CRT) before tumor tissue collection were more likely to be in the TMB-(6+) category but this trend did not reach significance (*p* = 0.09). Exposure to treatment with radiation or chemotherapy or both prior to tissue collection was not associated with TMB.

A strongly significant correlation was found between TMB and primary tumor location (*p* < 0.01). Specifically, the patients within the TMB-(6+) category were significantly more likely to have cancers of the larynx in comparison to other locations (72.7% vs. 38.9%; *p* < 0.01) and significantly less likely to have cancers of the oropharynx as opposed to other locations (29.4% vs. 70.6%; *p* < 0.01). The stage at the time of diagnosis (I–IV) did not correlate to TMB. Similarly, there was no correlation between T stage, N stage, or M stage and TMB.

### 3.5. Prognostic Value of PD-L1 Expression

For all patients with PD-L1 expression data, the median follow-up time for testing was 588 days from the time of diagnosis with a median survival time of 791 days (95% CI 708 to 1199 days). Overall, 47 patients (66%) were alive at one year from diagnosis and 13 (23.2%) were alive at two years from diagnosis. Survival at one or two years was not associated with PD-L1 expression level. More than half of surviving patients (67%) had residual disease at last visit. There was no correlation between PD-L1 groups and the extent of disease at last visit.

PD-L1 expression did not correlate to survival in 2tPD-L1 or 3tPD-L1 analysis from time of tissue collection or from time of diagnosis. In the 2tPD-L1 analysis, the median survival in the 2tPD-L1-(<20) group was 521 days (95% CI 412–1008 days) and was not significantly different from median survival in the 2tPD-L1-(20+) group, 541 days (95% CI 415–666 days) (*p* = 0.89). Survival from time of diagnosis was also found to be insignificantly different between 2tPD-L1 groups where the median survival in the 2tPD-L1-(<20) group was 1044 days (95% CI 711–1292 days) and the median survival in the 2tPD-L1-(20+) group was 752 days (95% CI 504–1320 days) (*p* = 0.47). Differences in survival between 3tPD-L1 groups were similarly unimpressive in regard to time from tissue collection and diagnosis (*p* = 0.95 and *p* = 0.51, respectively). Additional information regarding correlation between PD-L1 and outcome/survival can be found in Table 3, Table 4, and Figure 3. In an adjusted Cox proportional hazard regression model, the impact of the 3t-PD-L1 variable was not related to survival when controlled for age, smoking, nodal status, subsite, and exposure to CRT (*p* = 0.09) and related to poorer OS in the two-tiered analysis (*p* = 0.03) (Table 3 and Table 4). When TMB score was added to the adjusted variables above, the 2tPD-L1 more accurately could predict survival; however, this, too, did not quite reach significance (*p* = 0.051) (Table 4).

### 3.6. Prognostic Value of TMB

For all patients with TMB score data, the median follow-up time was 616 days from the time of cancer diagnosis with a median survival from diagnosis of 521 days (95% CI 412–1008 days). Overall, 72 patients (69.9%) were alive at one year from diagnosis, and 25 (30.86%) were alive at two years from diagnosis. At the time of the last visit, 39% of patients had no evidence of disease, 24% had recurrent or progressive locoregional disease, 12% had metastatic disease, and 25% had locoregional and metastatic disease. There was no association between disease status at last visit and TMB (*p* = 0.39).

Generally, patients within the TMB-(6+) category fared better than those within the TMB-(<6) category, but differences in survival at 1 and 2 years were not significant (Table 3). Survival from the time of diagnosis was significantly better in patients within TMB-(6+) (752 days (95% CI 599–905 days) and 1165 days (95% CI beginning at 902 with the upper limit not yet reached, *p* = 0.01). When considered from the time of tumor sample collection, survival was only marginally better in the TMB-(6+) category (*p* = 0.08). The survival significance from the time of diagnosis was upheld in an adjusted Cox proportional hazard regression model of survival controlled for PD-L1 category, age, smoking, nodal status, subsite, and previous exposure to CRT (*p* = 0.02). This remained significant in an analysis in which 2tPD-L1 was added to the analysis (*p* < 0.01) (Table 3, Table 4, and Figure 3).

### 3.7. Treatment with PD-L1/PD-1 Inhibitors and Correlation with PD-L1 and TMB

A total of 79 patients in this study received at least one treatment with an ICI. Treatment efficacy was able to be evaluated in 51 of these patients. Of the 28 patients that could not be evaluated, nine patients (32%) received a planned treatment with less than three administrations of a PD-L1 inhibitor in a neoadjuvant setting and were not able to be evaluated for treatment efficacy. Nineteen patients (68%) had treatment cessation before the initial scans to measure therapeutic response, due to treatment toxicity, poor tolerance, continued rapid progression of malignancy leading to complications or hospice transitions, or decision to discontinue treatment by the patient for other reasons. Of the 51 patients with measurable response, there were 27 patients who progressed, 3 patients with stable disease, 10 patients with partial response, and 11 patients with complete response. In total, 44 of the 51 patients were treated with pembolizumab, and only two patients were treated with a PD-L1 inhibitor within a clinical protocol (Appendix A). In all analyses regarding response to ICI, patients who had CR, PR, or SD with more than 6 months duration were grouped in a category referred as “responders” and were compared against patients with tumor progression referred to as “nonresponders”. Amongst all 51 patients, 24 patients were categorized as responders, and 27 patients were categorized as nonresponders. Responders received an average 19.6 (range: 3–35) administrations of an ICI, while nonresponders received only 5.3 (2–9) administrations. The average PFS was 910 days for patients with CR, 388 days for patients with PR, 237 days for the patients with SD, and 109 days for patients with PD. PFS was statistically significantly longer in responders with an average of 661.7 days (185 to 1825 days) vs. nonresponders who had an average of 109 days (63–190 days) (*p* < 0.01).

Of the 51 evaluable patients, there were 36 patients with available PD-L1 expression data with 19 patients defined as responders and 17 patients as nonresponders. There were no associations found between PD-L1 level and response to treatment with ICI. When compared as a continuous variable, the mean PD-L1 was 26.4 (95% CI 9.6–43.2) in responders vs. 26.5 (95% CI 11.0–41.9) in nonresponders (*p* = 0.99). When compared as a three-tiered variable, the percentage of responders and nonresponders in the 3tPD-L1-(0) category was 10.5% and 11.6%, respectively, and 36.8% and 41.1% in the 3tPD-L1-(20+) category (*p* = 0.89). Similarly, there were no associations identified between PD-L1 and PFS in patients treated with ICIs. PFS was 208 days for 3tPD-L1-(0), 374 days for 3tPD-L1-(1–19), and 404 days for the 3tPD-L1-(20+) category (*p* = 0.66). There was no significant correlation between PD-L1 measured as a continues variable and PFS (*p* = 0.62)

There were 40 patients with available TMB data that were able to be evaluated for treatment response, with 20 patients categorized as responders and 20 patients categorized as nonresponders. There was a statistically significant association between the response to treatment with ICI and continuous TMB, with a mean TMB of 11.3 mut/Mb (95% CI 6.6 mut/Mb-16.0 mut/Mb) in responders and 4.9 mut/Mb (95% CI 3.4 mut/Mb-6.4 mut/Mb) in nonresponders (*p* = 0.01). Correlation of treatment response with TMB as a categorical variable demonstrated a similar correlation, with 12 responders (60% of the total responders) and 6 nonresponders (30% of the total responders) found within the TMB-(6+) category (*p* = 0.056). Similarly, there was a statistically significant association of TMB with PFS when analyzed as a continues variable (*p* = 0.01), and statistical significance was maintained in a categorical analysis, with PFS found to be 261.7 days in the TMB-(0–5) and 538.7 in the TMB-(6+) (*p* = 0.04).

## 4. Discussion

The establishment of PD-L1 ICIs has brought about a new era in the management of patients with HNSCC. Efforts in clinical research are now focused on defining strategies to increase the efficacy of PD-L1 ICIs by identifying those who are best suited to receive these therapies. PD-L1 and TMB have been the dominant targets investigated as potential biomarkers of response to ICIs, yet, especially in HNSCC, the results remain not only scarce but frequently inconsistent.

In this single-institution retrospective analysis, the PD-L1 and TMB data of 132 HNSCC patients (95 patients with PD-L1 data, 128 patients with TMB data, and 91 patients with both PD-L1 and TMB data) were correlated with their demographics, survival and, when appropriate, response to ICI. This study population is consistent with a standard HNSCC population in terms of age, gender, race, smoking, and HPV status (Table 1). Conventional prognostication tools held true in this analysis. Smoking, as well as advanced nodal stage were associated with worse survival in a multivariate analysis (Table 4). Both PD-L1 and TMB were measured with standardized, commercially available methods. To the authors’ knowledge, this is one of few studies involving a comprehensive analysis of both TMB and PD-L1 in a dedicated HNSCC population, utilizing universally available, standardized measurements of each variable. Published studies present frequently conflicting results, most likely affected by variations in the utilized assays, as well as variations in the thresholds used to define results.

The median PD-L1 score in this study was 10%, and 87% of patients with PD-L1 data expressed PD-L1 positivity (PD-L1 expression ≥ 1). This proportion of PD-L1 positive disease is slightly higher than previous HNSCC cohorts analyzed in the literature where studies have demonstrated positivity rates between 57% and 82% [5,10,11]. It was thought that earlier disease stage might account for this finding, as significantly more patients in this study who were diagnosed at an earlier stage (overall and with respect to T stage alone) were found to have higher PD-L1 expression (Table 2), and there was a relative surplus of early-stage patients in the study cohort in comparison to the predominance of recurrent/metastatic disease in cohorts from the literature.

PD-L1 was evaluated as both a three-tiered and two-tiered variable. The tier cutoff points of the 3tPD-L1 variable were influenced by the KEYNOTE-048 study [32]. Although the three tiered approach had the considerable advantage of distinguishing PD-L1 negative disease (3tPD-L1-(0)), an important category in the decision tree guiding therapy in the current standard first line management of metastatic/recurrent HNSCC, the distribution of the patients in the resultant groups were dissimilar. In an attempt to help offset the bias introduced by the lack of uniformity in the methods used to measure PD-L1 expression (TPS vs. CPS) and the low number of patients in the 3tPD-L1-(0) group, the decision was made to combine the lower two categories into a single group, thereby generating the two-tiered variable (2tPD-L1). Ultimately, there were more significant associations between the 3tPD-L1 variable and demographic data, thus illustrating the importance of distinguishing PD-L1 positivity from PD-L1 negativity in HNSCC.

The median TMB analyzed was 5 mut/Mb, consistent with other reports [33]. Although the split of TMB between groups was based on Foundation Medicine guideline, the groups would be similar for a threshold based on median TMB, with 70 patients in the TMB-(<6) group and 58 patients in the TMB-(6+) group. Similar to the reports in HNSCC [27,28] and in other malignancies [24,33,34], this study demonstrated that there was no significant relationship between PD-L1 and TMB, reflecting the dynamic interactions between the two variables.

This study is the first to demonstrate an association between higher PD-L1 expression and the African American race. This held true in both the 2tPD-L1 and the 3tPD-L1 analysis. Notably, the 3tPD-L1 analysis also correlated Caucasian race to lower PD-L1 in a statistically significant manner. Conversely, a higher proportion of Caucasians had high TMB-(6+) than African Americans (48.1% vs. 23%), although this higher proportion was not statistically significant.

A study of The Cancer Genome Atlas (TCGA) HNSCC population did not find any significant correlation between TMB and race or gender but did identify a statistically significant association between high TMB and age above 60 [34]. This finding was confirmed by other reports on the same public TCGA HNSCC database [35] as well as in another study involving 100,000 human cancer genomes in which many malignancies were considered [33]. Such correlations of TMB or PD-L1 with age were not identified in this study.

Female gender was the only category in this analysis to show a trend toward both higher PD-L1 expression (48% of females vs. 35.3% of males were in the 3tPD-L1-(20+) group) and higher TMB (53.8% of females vs. 41.6% of males were in the TMB-(6+) group). This trend did not reach statistical significance possibly due to the inherent lower number of females inflicted by this type of cancer and subsequently being included in our analysis. Similar association of PD-L1 overexpression with female gender was reported by two studies in patients with oral cavity SCC [36,37], while a retrospective study of patients with oropharyngeal HNSCC reported no significant difference in age and gender and PD-L1 expression [38].

Smokers were found to have both a lower level of PD-L1 expression and higher TMB, compared with never-smokers or former smokers. This TMB finding is further supported by two other recent reports. One such project was dedicated to a population size similar to this study [28] and another analyzed the TCGA-HNSCC [34]. Our study additionally demonstrated an association between alcohol consumption and lower PD-L1 expression. A review of the literature revealed two studies, focused on oral cavity SCC on this topic (a meta-analysis [36] and a smaller retrospective study of 55 patients [39]) which found a similar significant relationship between low-PD-L1 with alcohol consumption.

There are no consensus data regarding HPV status in association with PD-L1 and TMB. Previous studies in HNSCC concerning patients treated with surgery and adjuvant CRT [40] and those with oropharyngeal primaries [41] reported a significant correlation of PD-L1 expression with p16 status. Conversely, another study regarding patients with oropharyngeal SCC did not identify any significant difference in PD-L1 expression between HPV positive and negative tumors [38]. Similarly, there are conflicting data in the literature concerning the relationship between HPV/p16 and TMB. Some studies have shown that high TMB was associated with HPV negative disease [28,42], and others have demonstrated no significant correlation between the two variables [34]. There was no significant association between HPV and PD-L1 or TMB in this study. It was noted, however, that more patients with HPV negative disease were in the 3tPD-L1-(0) group (18.5% of patients), and no patients with 3tPD-L1-(0) had HPV positive disease. Although these findings did not reach significance, they support the theory that the relationships between PD-L1, TMB, and immune cell infiltration are more complicated and the immune pathways that assist in response are influenced by many components of the tumor microenvironment including HPV status, alcohol use, and/or tobacco use [42].

In this cohort, a trend that did not reach statistical significance suggested that BMI greater than 30 was associated with high PD-L1 and low TMB. Interestingly, reports in 976 patients with diverse tumors treated with PD-1/PD-L1 inhibitors showed that the response to treatment was significantly higher in overweight/obese patients compared to nonoverweight patients [43].

In concordance with two other reports on patients with HNSCC [34,44], this study found that TMB was associated with tumor location in a statistically significant way. The proportion of patients with laryngeal tumors was significantly increased, and oropharyngeal cancer decreased, in the TMB-(6+) group when compared to patients with any other throat tumor location. No correlation of PD-L1 with tumor location was identified.

Advanced cancer stage (I–IV) and advanced T stage (T3–T4 vs. T0–T2) were associated with low PD-L1 in this analysis. Of the 12 patients with PD-L1(0), seven patients had T4, and five patients had T3 tumors (Table 4). This finding is supported by a meta-analysis in patients with oral cavity SCC [36] but diverges from a study in oropharyngeal cancer [38]. Furthermore, in a review of a TCGA HNSCC population, Zhang et al. reported an association of advanced clinical stage and large tumor size with TMB rather than with PD-L1 [42]. This correlation was not identified in our study.

PD-L1 did not correlate with survival at 1 or 2 years or with OS in univariate analysis (Table 3). In the multivariate survival analysis of the 2tPD-L1 variable, 2tPD-L1-(20+) predicted worse survival comparative with 2tPD-L1-(0–19). Previous studies have found the same, including one meta-analysis and two retrospective reviews, all addressing oral cavity SCC patients [36,37,45]. Another meta-analysis in HNSCC and a study in oropharyngeal SCC patients reported no association of PDL1 expression with survival [38,46]. Conversely, three retrospective reviews of HNSCC patients reported the association of PD-L1 expression with improved OS [40,47,48,49].

TMB significantly correlated with OS measured from the time of diagnosis (Table 3). This significance was maintained in a Cox proportional hazards regression model when adjusted for age, tobacco use, tumor site, nodal stage at diagnosis, previous treatment with chemotherapy, radiation or combined chemoradiation therapy, and PD-L1 level in a multivariate analysis model (Table 4). Similar with PD-L1, the literature reports are controversial regarding TMB’s association with survival. This is not surprising given the expected influences of disease characteristics, treatment, biopsy sites, and the variability in measurement techniques. Additionally, the finding that those in the higher TMB group had a better response to PD-L1 ICI in combination with the facts that a high proportion of our patient population had TMB-(6+) and were treated with PD-L1 ICI, most likely influenced survival in this study. Reports in the literature support the correlation between high TMB and improved OS, including the reports of univariate and multivariate survival of patients with oral cavity squamous cell cancer treated with surgery as their primary intervention [50]. Conversely, a multicenter retrospective study of patients treated with definitive CRT found a significant correlation of TMB with poor survival [51]. Finally, a study of 10,000 patients from TCGA with different tumors showed an association of TMB with response to IO but not with OS [25].

A total of 79 patients in this study received at least one treatment with an ICI, of which 51 patients were evaluable for treatment response. It should be noted that the percentage of responders in this study is higher than previously reported in the literature (26.5%), with a particularly high percentage of patients with CR (13.9%). This finding might be correlated with the fact that almost half of these patients (5 out of 11 patients) were treated with other therapeutic interventions that might have potentiate immune response to ICIs (palliative radiotherapy (three patients), combined palliative chemotherapy (one patient), and concurrent definitive chemoradiotherapy for a second head and neck cancer primary (one patient)) (Appendix A). One patient with metastatic HNSCC who achieved a durable CR (1875 days to date, with no recurrence) after just three administrations of a PD-1 inhibitor will be presented in a separate publication. Of the 51 patients with evaluable response, 40 patients had TMB results, and 36 patients had PD-L1 results. There was a statistically significant association between the response to treatment with ICI and continuous TMB score with a mean TMB of 11.2 in responders and 4.9 in nonresponders (*p* = 0.01). Evaluation as a categorical variable demonstrated that 66.6% of the responders and 33.3% of the nonresponders were within the high TMB (6+) category (*p* = 0.055). Furthermore, TMB corelated significantly with PFS in both categorical and continuous analysis. There are other published reports supporting TMB as a possible predictor of response to ICIs. In a retrospective analysis of 126 HNSCC patients treated with anti-PD-1/PD-L1 agents, TMB was found to be significantly higher among responders (21.3 vs. 8.2 mut/MB, *p* < 0.01) [28]. The study of a cohort from KEYNOTE-012 sought to characterize this further and demonstrated that TMB was predictive of response to pembrolizumab in HPV negative patients but not in HPV positive patients [29]. Finally, though the role of circulating/blood TMB has yet to be defined, retrospective studies in HNSCC have a linked response to ICI with circulating/blood TMB ≥ 16 mut/Mb [45,47].

There was no association between response to treatment with ICI or PFS and PD-L1 level analyzed as a categorical (*p* = 0.66 and *p* = 0.89, respectively) or continuous variable (*p* = 0.62 and *p*= 0.99, respectively). In this study, PD-L1 values were measured by both TPS (55% of patients) and CPS (45% of patients), and due to the small sample size, no attempts were made to separate the analysis of PD-L1 by the reporting technique and correlate each of these distinct groups with PFS or response to treatment with ICI. It should be noted that the literature suggests that such an analysis could yield a different result; KEYNOTE-040 and -048 reported PD-L1 by CPS and demonstrated a significant correlation between PD-L1 and response to ICI [10,52,53], but CHECKMATE-141 failed to show a significant correlation between tumor response to Nivolumab and PD-L1 overexpression when PD-L1 was reported by TPS [5].

In summary, this study reported significant association of high PD-L1 expression with the African American race, nonsmoking and nonalcohol use, with early clinical cancer stage and early tumor stage, and with poor survival in a multivariate analysis. No predictive value for PFS or for BOR to ICIs was identified in the PD-L1 analysis. High TMB was reported to be significantly associated with smoking, tumor location in the larynx, and survival in both univariate and a multivariate analysis, as well as with PFS and BOR to ICIs.

Notably, this study comprehensively analyzed both PD-L1 and TMB in a dedicated HNSCC cohort. The utilization of standardized, commercially available methodologies is another unique feature among reports in HNSCC, encouraging the reproducibility and building of a consistent database. A direct comparison between TMB and PD-L1 results was not employed due to the variation of PD-L1 reporting (TPS and CPS) triggered by the more recent approval by the FDA of CPS as a companion diagnostic. The other limitations of this study include the retrospective nature of the review and a limited sample size, especially in the analysis regarding response to ICI.

### Future Directions

Furthermore, additional studies are needed to generate the necessary context and framework of standardized variables aimed to predict response of HNSCC to IO in general and to ICIs in particular. The standardization of assays is the next step in assisting with the creation of consistent results and the development of thresholds for high and low scoring groups that are both sensitive and specific in HNSCC for further predictive analysis. The recent availability of TMB as a circulating biomarker that bypasses the need for tissue procurement and allows a dynamic assessment makes it a more attractive biomarker. The association of TMB with prognosis and response to ICI presented by this study and others warrants further attention and prompts the advancement of TMB in future prospective clinical studies of ICIs, with the ultimate goal of becoming a companion diagnostic for recommendation of ICIs in HNSCC.

## 5. Conclusions

ICIs have changed the landscape of the treatment of HNSCC. Regardless, less than 20% of the treated patients benefit from these novel therapeutics, prompting urgent studies to help identify predictors of response and improve patient selection. This study has demonstrated the utility of TMB as a prognostic variable and predictive marker of response to ICI. In addition, the study pointed to the significant association of high TMB with active tobacco use and with primary tumor location in the larynx. High PD-L1 values were associated with the African American race, high T stage, high overall disease stage, non-/ex-smokers, and non-/ex-drinkers. More information is needed to create a framework in which PD-L1 and TMB co-exist with other variables to predict response to ICI on an in-dividual level. Nonetheless, the existing data for each of these independent variables are promising in the world of precision oncology, and the results of the current study argue for the advancement of TMB in prospective research.

## Figures and Tables

**Figure 1 cancers-13-05733-f001:**
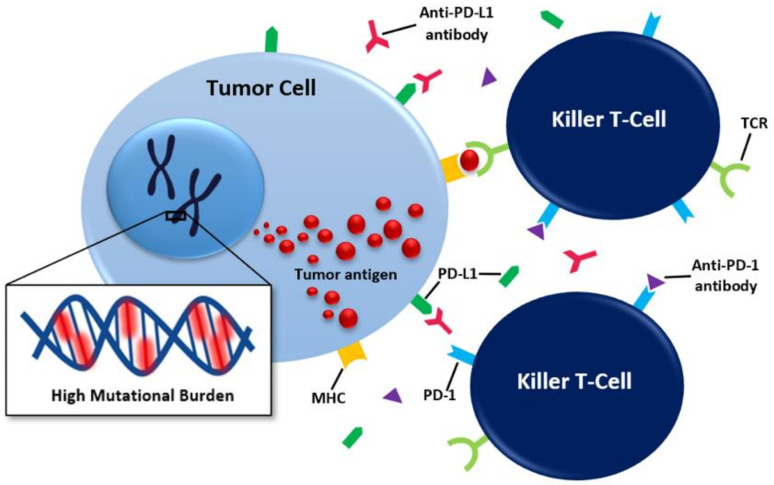
A brief review of the mechanism of action of PD-1/PD-L1 ICIs. Abbreviations: MHC, major histocompatibility complex; PD-1, programmed death-1; PD-L1, programmed death ligand-1; and TCR, T-cell receptor.

**Figure 2 cancers-13-05733-f002:**
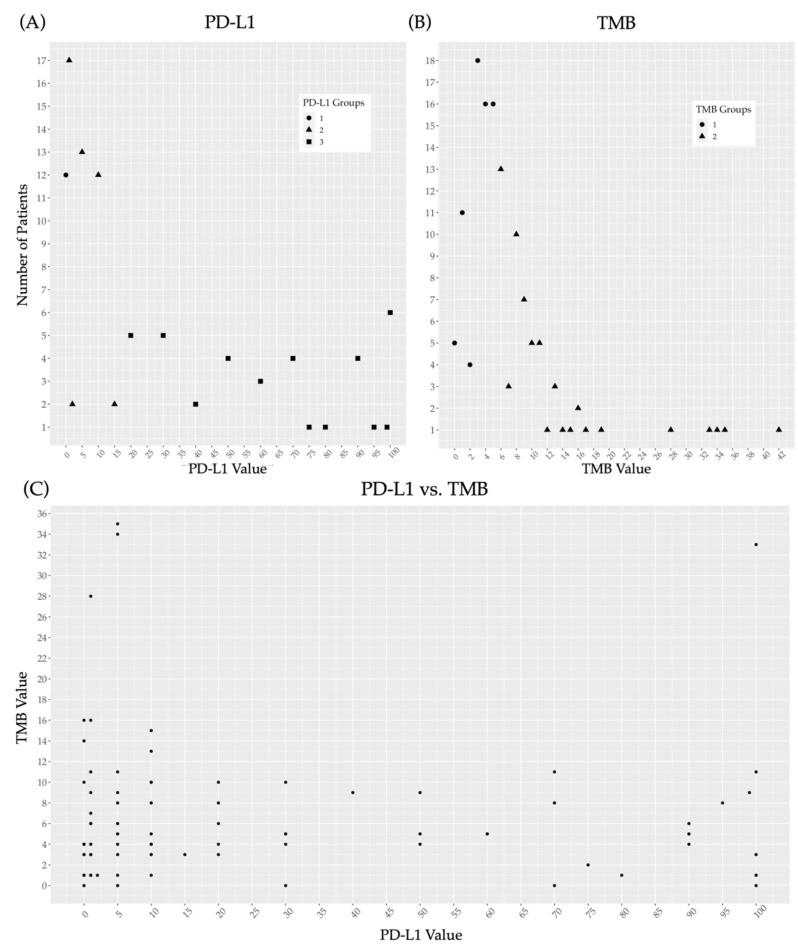
Distribution of continuous PD-L1 and continuous TMB and scatter plot demonstrating the relationship between each. (**A**) Distribution of PD-L1 across 3tPD-L1-(0) (group 1, circles), 3tPD-L1-(1–19) (group 2, triangles) and 3tPD-L1-(20+) (group 3, squares); (**B**) Distribution of TMB across TMB-(<6) (group 1, circles) and TMB-(6+) (group 2, triangles); (**C**) Scatter plot demonstrating the failure of PD-L1 to correlate to TMB. Abbreviations: PD-L1, programmed death ligand-1; and TMB, tumor mutational burden.

**Figure 3 cancers-13-05733-f003:**
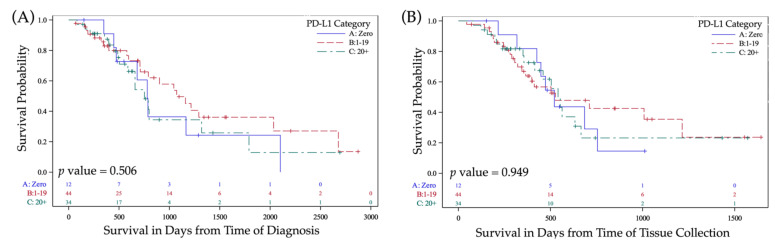
Kaplan –Meier curves for PD-L1 and TMB variables. (**A**) Survival from time of diagnosis in patients with 3tPD-L1-(20+) vs. 3tPD-L1-(1–19) vs. 3tPD-L1-(0); (**B**) Survival from time of tissue acquisition in patients with 3tPD-L1-(20+) vs. 3tPD-L1-(1–19) vs. 3tPD-L1-(0); (**C**) Survival from time of diagnosis in patients with TMB-(6+) vs. TMB-(<6); (**D**) Survival from time of tissue acquisition in patients with TMB-(6+) vs. TMB-(<6). Abbreviations: PD-L1, programmed death ligand-1; 3tPD-L1-(20+), PD-L1 ≥ 20; 3tPD-L1-(1–19), 0 < PD-L1 < 20; 3tPD-L1-(0), PD-L1=0; and TMB, tumor mutational burden; TMB-(<6), TMB less than 6; TMB-(6+), TMB greater than or equal to 6. Legend: Blue solid lines indicate survival curves for patients with TMB-(<6) or with 3tPD-L1-(0); Red dashed lines indicate survival curves for patients with TMB-(6+) or with 3tPD-L1-(1–19); Green dashed lines indicate curves for patients with 3tPD-L1-(20+) in the 3tPD-L1 analysis. Abbreviations: PD-L1, programmed death-ligand1; and TMB, tumor mutational burden.

**Table 1 cancers-13-05733-t001:** Characteristics of all patients included in the study.

Characteristics	TMB Patients	PD-L1 Patients	Characteristics	TMB Patients	PD-L1 Patients
	No. (%)	No. (%)		No. (%)	No. (%)
**Age at Diagnosis (Years)**			**Primary Tumor Location**		
Median	60	61	Nasopharynx	8 (6.2)	7 (7.4)
≥60	64 (50)	50 (52.6)	Oropharynx	50 (39.1)	37 (38.9)
<60	64 (50)	45 (47.4)	Oral Cavity	33 (25.8)	23 (24.2)
			Hypopharynx	7 (5.5)	5 (5.3)
**Gender**			Larynx	22 (17.2)	16 (16.8)
Male	89 (69.5)	68 (71.6)	Sino-Nasal	5 (3.9)	4 (4.2)
Female	39 (30.4)	27 (28.4)	Unknown	3 (2.3)	3 (3.2)
**Race**			**Disease Stage at Time of Diagnosis**		
Caucasian	108 (84.4)	79 (83.2)			
African American	13 (10.2)	13 (13.7)			
Other	7 (5.4)	3 (3.1)	**Cancer Stage**		
			I	19 (14.8)	17 (17.9)
**ETOH Status**			II	21 (16.4)	18 (18.9)
Never	63 (49.2)	46 (48.4)	III	29 (22.7)	15 (15.8)
Former	28 (21.9)	22 (23.2)	IV	59 (46.1)	45 (47.4)
Active	37 (28.9	27 (28.4)			
			**Cancer Stage IV**		
**Smoking Status**			IVA	39 (66.1)	29 (64.4)
Never	37 (28.9)	29 (30.5)	IVB	14 (23.7)	11 (24.4)
Former	39 (30.5)	30 (31.6)	IVC	6 (10.2)	5 (11.2)
Active	52 (40.6)	36 (37.9)			
			**N Stage**		
**HPV and/or p16**			N0	37 (28.9)	27 (28.4)
Negative	56 (43.8)	43 (45.3)	N1	29 (22.7)	20 (21.1)
Positive	40 (31.2)	29 (30.5)	N2	49 (38.3)	37 (38.9)
Not Tested	32 (25)	23 (24.2)	N3	13 (10.1)	11 (11.6)
**BMI**			**Tissue**		
			**Source**		
<18.5	19 (14.7)	11 (11.9)	Primary Tumor	83 (65.4)	55 (60.4)
18.5–24.9	42 (32.6)	33 (35.9)	Regional Node	11 (8.7)	10 (11.0)
25–29.9	39 (30.2)	30 (32.6)	Metastatic Lesion	11 (8.7)	9 (9.9)
≥30	29 (22.5)	18 (19.6)	Recurrence	22 (17.3)	17 (18.7)

Abbreviations: BMI, body mass index; HPV, human papilloma virus; PD-L1, programmed death ligand-1; TMB, tumor mutational burden.

**Table 2 cancers-13-05733-t002:** Correlation of select demographic and patient specific data with 3-tiered-PD-L1 and 2-tiered TMB categories.

Variable		3tPD-L1 Analysis				TMB Analysis		
		No. (Row %)			*p*	No. (Row %)		*p*
		≥20	1–19	0		≥6	<6	
**Age**	<60	14 (31.1)	25 (55.6)	6 (13.3)	0.240	31 (48.4)	33 (51.6)	0.478
	≥60	23 (46.0)	21 (42.0)	6 (12.0)		27 (42.2)	37 (57.8)	
**Gender**	Male	24 (35.3)	35 (51.5)	9 (13.2)	0.510	37 (41.6)	52 (58.4)	0.199
	Female	13 (48.1)	11 (40.7)	3 (11.1)		21 (53.8)	18 (46.2)	
**Race**	Caucasian	28 (35.4)	41 (51.9)	10 (12.7)	** *0.046* **	52 (48.1)	56 (51.9)	* 0.087 *
	AA	9 (69.2)	2 (15.4)	2 (15.4)		3 (23.1)	10 (76.9)	
**HPV**	+	11 (38.0)	18 (62.1)	0 (0.0)	0.354	16 (40.0)	24 (60.0)	0.804
	−	18 (41.9)	17 (39.5)	8 (18.6)		21 (37.5)	35 (62.5)	
**Smoking** **History**	Current	12 (33.3)	17 (47.2)	7 (19.4)	** *0.044* **	31 (59.6)	21 (40.4)	** *0.029* **
	Former	10 (33.3)	16 (53.3)	4 (13.3)		13 (33.3)	26 (66.7)	
	Never	15 (51.7)	13 (44.8)	1 (3.4)		14 (37.8)	23 (62.2)	
**Alcohol** **History**	Current	7 (25.9)	15 (55.6)	5 (18.5)	** *0.012* **	18 (48.6)	19 (51.4)	0.326
	Former	6 (27.3)	12 (54.5)	4 (18.2)		15 (53.6)	13 (46.4)	
	Never	24 (52.2)	19 (41.3)	3 (6.5)		25 (39.7)	38 (60.3)	
**BMI**	≥30	10 (55.5)	7 (38.9)	1 (5.6)	* 0.072 *	9 (31.0)	20 (69.0)	* 0.071 *
	<30	25 (33.8)	37 (50.0)	12 (16.2)		50 (50.0)	50 (50.0)	
**Previous** **CRT**	+	15 (41.7)	16 (44.4)	5 (13.9)	0.868	23 (51.1)	22 (48.9)	0.331
	−	22 (37.3)	30 (50.8)	7 (11.9)		35 (42.2)	48 (57.8)	
**Primary** **Tumor** **Location**	OC	10 (43.5)	11 (47.8)	2 (8.7)	0.400	16 (48.5)	17 (51.5)	0.685
	Other	21 (36.2)	28 (48.3)	9 (15.5)		35 (44.3)	44 (55.7)	
	Laryngeal	6 (37.5)	8 (50)	2 (12.5)	0.984	16 (72.7)	6 (27.3)	** *0.004* **
	OTT	25 (38.5)	31 (47.7)	9 (13.8)		35 (38.9)	55 (61.1)	
	OP	15 (40.5)	16 (43.2)	6 (16.2)	0.965	15 (30.0)	35 (70.0)	** *0.003* **
	Other	16 (36.4)	23 (52.3)	5 (11.4)		36 (58.1)	26 (41.9)	
**Stage at** **Diagnosis**	I	10 (58.8)	7 (41.2)	0 (0.0)	** *0.035* **	7 (36.8)	12 (63.2)	0.522
	II	7 (38.9)	11 (61.1)	0 (0.0)		11 (52.4)	10 (45.5)	
	III	4 (26.7)	6 (40.0)	5 (33.3)		11 (37.9)	18 (62.1)	
	IV	16 (35.6)	22 (48.9)	7 (15.6)		29 (49.2)	30 (50.8)	
	IVA	5 (71.4)	2 (28.6)	0 (0.0)	0.802	22 (56.4)	17 (43.6)	0.146
	IVB	13 (59.1)	6 (27.3)	3 (13.6)		5 (35.7)	9 (64.3)	
	IVC	11 (68.8)	3 (18.8)	2 (12.5)		2 (33.3)	4 (66.7)	
	T0–2	20 (46.5)	23 (53.5)	0 (0.0)	** *0.008* **	23 (42.6)	31 (57.4)	0.864
	T3–4	17 (54.8)	2 (6.5)	12 (38.7)		39 (52.7)	35 (47.3)	
	N0	11 (40.7)	10 (37.0)	6 (19.4)	0.949	17 (45.9)	20 (54.1)	0.751
	N1	9 (45.0)	11 (55.0)	0 (0.0)		11 (37.9)	18 (62.1)	
	N2	13 (35.1)	20 (54.0)	4 (10.8)		24 (49.0)	25 (51.0)	
	N3	4 (36.3)	5 (45.5)	2 (18.2)		6 (46.2)	7 (53.8)	
	M0	34 (38.2)	43 (48.3)	12 (13.5)	0.372	54 (45.0)	66 (55.0)	0.784
	M1	3 (50.0)	3 (50.0)	0 (0.0)		4 (50.0)	4 (50.0)	

Results with *p* ≤ 0.05 are bolded in italics; results with 0.05 < *p* < 0.10 are italicized and underlined. 3tPD-L1, 3-tiered PD-L1; AA, African American; BMI, Body Mass Index; CRT, chemoradiotherapy; HPV, human papilloma virus; OC, oral cavity; OP, oropharyngeal; OTT, other throat tumors; PD-L1, programmed death ligand-1; and TMB, tumor mutational burden.

**Table 3 cancers-13-05733-t003:** Association of high PD-L1 expression and TMB with survival outcomes.

Survival Start Time Point	Overall Survival Univariate Analysis for Highest Scores			Overall Survival Adjusted Analysis for Highest Scorers			1 Year OS	2 Year OS
	HR	95% CI	*p* value	HR	95% CI	*p* value	*p* values	
**2tPD-L1**								
From Time of Diagnosis	1.28	0.72–2.27	0.473	2.02	(1.06–3.86)	** * 0.033 * **	0.951	0.320
From Time of Sample Collection	1.05	0.57–1.97	0.888					
**3tPD-L1**								
From Time of Diagnosis	0.97	0.42–2.22	0.938	1.28	(0.46–3.61)	* 0.092 *	0.522	0.386
From Time of Sample Collection	0.95	0.41–2.24	0.949					
**TMB**								
From Time of Diagnosis	0.51	0.30–0.86	* 0.081 *	0.49	(0.27–0.90)	** * 0.021 * **	0.950	0.121
From Time of Sample Collection	0.63	0.37–1.06	** * 0.014 * **					

Results with *p* < 0.05 are bolded in italics and underlined; results with 0.05 < *p* < 0.10 are italicized and underlined. Abbreviations: 3tPD-L1, 3-tiered PD-L1; 2tPD-L1, 2-tiered PD-L1; CI, confidence interval; HR, hazard ratio; PD-L1, programmed death ligand-1; OS, overall survival; and TMB, tumor mutational burden.

**Table 4 cancers-13-05733-t004:** Results from adjusted Cox proportional hazard regression models of impact of the presence of PD-L1 and TMB on overall survival.

	3tPD-L1			TMB			TMB and 2tPD-L1					
	HR	95% CI	*p*	HR	95% CI	*p*	PD-L1 HR	PD-L1 95% CI	PD-L1 *p*	TMB HR	TMB 95% CI	TMB *p*
**Highest Scoring** **Groups**	1.28	(0.46–3.61)	* 0.092 *	0.49	(0.27–0.90)	** * 0.021 * **	1.99	(1.0–3.97)	* 0.051 *	0.35	(0.16–0.76)	** * 0.008 * **
	***p* values for Adjusted** **Variables**			***p* values for Adjusted** **Variables**			***p* values for Adjusted** **Variables**					
**Age Below 60 Years Old** (yes vs. no)	0.52			0.508			0.656					
**Smoking** (never vs. ever)	** * 0.016 * **			* 0.058 *			** * 0.005 * **					
**N Stage** (N0, N1, N2, N3)	** * 0.003 * **			0.709			** * 0.007 * **					
**Subsite** (OP vs. OC vs Pharynx vs. other)	0.338			0.126			0.518					
**Previous CRT** (yes vs. no)	** * 0.027 * **			** * 0.0209 * **			0.182					

Analyses with *p* < 0.05 are bolded in italics and underlined; results with 0.05 < *p* < 0.10 are italicized and underlined; Abbreviations: 2tPD-L1, 2-tiered PD-L1; 3tPD-L1, 3-tiered PD-L1; CI, confidence interval; HR, hazard ratio; N/A, nonapplicable; CRT, combined chemotherapy and radiation therapy; PD-L1, programmed death ligand-1; and TMB, tumor mutational burden.

## Data Availability

The datasets analyzed are available from the corresponding author upon request.

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
