# Peer review of "Relationship between Tumor Mutational Burden, PD-L1, Patient Characteristics, and Response to Immune Checkpoint Inhibitors in Head and Neck Squamous Cell Carcinoma"

_cancers, 2021, doi:10.3390/cancers13225733_

Round 1

Reviewer 1 Report

Dear Authors,

The article: 'Relationship Between Tumor Mutational Burden, PD-L1, Patient Characteristics and Response to Immune Checkpoint Inhibitors in Head and Neck Squamous Cell Carcinoma' was to investigate the feasibility of continued pursuits of PD-L1 and TMB in prospective clinical trials in which ICI would be used to treat HNSCC.

English language and style are fine.

Punctuation mistakes should be corrected. p value should be written italics.

The article is well planned and prepared. It contains a decent summary of the analyzed topic.

To sum up, article can be accepted after minor revision.

Author Response

Reviewer 1,

Thank you for your very kind review. You will find that all of the recommendations you have made have been addressed. More improvements to grammar and syntax have been made throughout the text and are available for your review via the track changes feature.

Specifically, the "p" in p value was italicized throughout the text. Once corrected, the tracked changes were accepted due to the significant burden of marking notes, making the manuscript difficult to follow.

Sincerely, 

Kimberly Burcher and colleagues 

Reviewer 2 Report

In this study, tumor mutational burden and expression of PD-L1 have been correlated with clinical data, patient characteristics, and response to check point inhibitors of patients with HNSCC. The authors could not detect any significant association between response to immunotherapy and PD-L1 expression or TMB.

Regarding immunohistochemical studies, several points are missed:
1) the exact protocols on applied method are missing.
2) a table of the determined values of the patients divided into CPS and TPS would be helpful.
3) representative immunohistological images of PD-L1 should be included
4) The CPS of 1 refers to the administration of pembrolizumab. Did all patients receive pebrolizumab? A detailed account of immunotherapy is missing here .

Author Response

Reviewer 2,

Thank you for the thorough review and for the useful comments and suggestions that helped improve the quality of our manuscript. You will find that all of the recommendations you have made have been addressed. More improvements to grammar and syntax have been made throughout the text and are available for your review via the track changes feature. Please find a specific list of critiques you raised listed below. Please not that several points regarded the immunohistochemical studies could not be addressed.

  1. The exact protocols on applied method are missing.
    • We added more information to better describe the method utilized for the analysis and reporting of PD-L1 results (lines 132 through 135). However, this is a standard FDA approved method performed in the commercial laboratories of Foundation Medicine and Mayo Clinic and further details of the protocols can be found on the Foundation Medicine Site.
  2. Representative immunohistological images of PD-L1 should be included.
    • The testing of PD-L1 was not performed in house, therefore IHC images are not available. The testing was performed by the commercial laboratories of Foundation Medicine and the Mayo Clinic by the standardized FDA approved method. Pictures are not provided with the standardized results.
  3. A table of the determined values of the patients divided into CPS and TPS would be helpful
    • We agree, and there have added Supplemental Table 1 to address this.
  4. The CPS of 1 refers to the administration of pembrolizumab. Did all patients receive pebrolizumab? A detailed account of immunotherapy is missing here
    • Thank you for the very helpful comments. As a result of this critique, we further reviewed our data and therefore created Supplemental Table 1 to present a “detailed account of immunotherapy” and the questions above about TPS and CPS, types of ICI agents utilized, number or treatments, other therapeutic interventions during the treatment with ICI, duration of treatment. Information and references in the text were added in Results (lines 577-588, 596-599, 608-611) and in the Discussion (lines 810-820). We also added the analysis of Progression-free survival and its correlation with PD-L1 and TMB that we thought would increase the account of immunotherapy and enhance the correlative analysis of the treatment predictive value of PD-l1 and TMB. Please see additional details in Methods (lines 180-18), Statistical Methods (lines 199-215), and Abstract (line 40).

In the process of detailed review of the treatment with ICI and analysis performed, we detected an error in importing results in Tables 3 and 4. Results from an older analysis were entered in columns 1 and 2, while the correct results were entered in column 3. The difference between the two analyses derived from updating the Previous CRT results (last line in the table). The updated analysis was conducted before the initial manuscript submission but the older results were mistakenly imported in columns 1 and 2. Patient population being the same, no statistical significance was affected. Results are now corrected in Tables 3 and 4 and throughout the Results section.

In addition, we...

  • Added explanation that the * sign for the first two authors means equal participation to this manuscript’ production
  • Added explanation that ^ means corresponding author
  • Updated information for several authors.
  • Updated authorship participation (lines 628 to 634).

Thank you again for your time and thoughtful consideration.

Sincerely, 

Kimberly Burcher and colleagues

Round 2

Reviewer 2 Report

Thank for the revision. It is now a good manuscript